# Preparation of Collagen/Hydroxyapatite Composites Using the Alternate Immersion Method and Evaluation of the Cranial Bone-Forming Capability of Composites Complexed with Acidic Gelatin and b-FGF

**DOI:** 10.3390/ma15248802

**Published:** 2022-12-09

**Authors:** Miki Hoshi, Masayuki Taira, Tomofumi Sawada, Yuki Hachinohe, Wataru Hatakeyama, Kyoko Takafuji, Shinji Tekemoto, Hisatomo Kondo

**Affiliations:** 1Department of Prosthodontics and Oral Implantology, School of Dentistry, Iwate Medical University, 19-1 Uchimaru, Morioka 020-8505, Japan; 2Department of Biomedical Engineering, Iwate Medical University, 1-1-1 Idaidori, Yahaba-cho 028-3694, Japan

**Keywords:** collagen/hydroxyapatite composite, acidic-gelatin, alternate immersion method, hydroxyapatite precipitation, basic-fibroblast growth factor, bone formation, bone histology, bone substitute materials, dental implantology

## Abstract

Bone-substitute materials are essential in dental implantology. We prepared collagen (Col)/hydroxyapatite (Hap)/acidic gelatin (AG)/basic fibroblast growth factor (b-FGF) constructs with enhanced bone-forming capability. The Col/Hap apatite composites were prepared by immersing Col sponges alternately in calcium and phosphate ion solutions five times, for 20 and 60 min, respectively. Then, the sponges were heated to 56 °C for 48 h. Scanning electron microscopy/energy-dispersive X-ray spectroscopy, Fourier-transform infrared spectroscopy, and X-ray diffraction analyses showed that the Col/Hap composites contained poorly crystalline Hap precipitates on the Col matrix. Col/Hap composite granules were infiltrated by AG, freeze-dried, and immersed in b-FGF solution. The wet quaternary constructs were implanted in rat cranial bone defects for 8 weeks, followed by soft X-ray measurements and histological analysis. Animal studies have shown that the constructs moderately increase bone formation in cranial bone defects. We found that an alternate immersion time of 20 min led to the greatest bone formation (*p* < 0.05). Constructs placed inside defects slightly extend the preexisting bone from the defect edges and lead to the formation of small island-like bones inside the defect, followed by disappearance of the constructs. The combined use of Col, Hap, AG, and b-FGF might bring about novel bone-forming biomaterials.

## 1. Introduction

Animal-derived collagen (Col) has several medical uses, including in artificial skin [1], vascular grafts [2], and nerve guides [3], due to its excellent biocompatibility, biodegradability, and widespread availability [4]. Col is a biopolymer and a major constituent of the extracellular matrix of several important tissues, such as skin and bone [5]. While in total, 29 types of Col are present in the body [6], type I Col is preferred in tissue engineering because of its fibrillar elaboration in triple-helical polypeptide chains, which result in favorable mechanical properties [6,7,8]. Although the use of conventional Col is considered old-fashioned, new devices using Col have recently been developed in tissue engineering [6,9,10], drug delivery systems [11,12,13], and three-dimensional-printed organs [7]. Therefore, Col-based biomaterials may have multiple clinical applications in the future.

Cross-linking of Col using physical or chemical methods is frequently performed to improve its durability and strength. The cross-linking method is outlined in Appendix A: Cross-Linking Method of Collagen (Col) (Appendix A) [14,15,16,17]. The techniques related to this study were physical de-hydrothermal treatment (DHT) and chemical cross-link by ethylene glycol diglycidyl ether (EGDE) or glutaraldehyde (GTA).

Col has also been combined with osteo-conductive hydroxyapatite (Hap) to produce composites used for bone regeneration in dental implantology and oral surgery [18,19,20,21]. We previously reported successful preparation of three bone-substitute Col and Hap materials using lyophilization, DHA and GTA cross-linking, alternate immersion for Hap precipitation, and Newton press [17]; lyophilization, DHT cross-linking, and blending of micro-sized inter-connected porous Hap [22]; and lyophilization, DHT cross-linking, addition of nano-sized Hap, and Newton press [23].

To further improve the bone-forming capability of materials, we decided to add the growth factor system to Col/Hap composites [24,25,26,27,28]. Three elements in tissue engineering, including growth factor, are briefly explained in Appendix A: Tissue Engineering and Therapeutic Effects of b-FGF (Appendix A) [28,29,30,31,32,33,34]. In Japan, use of basic fibroblast growth factor (b-FGF) as a growth factor is common and has been approved by the government [35], whereas bone morphogenic protein is no longer used. The therapeutic effect of b-FGF is also illustrated in Appendix A [28,29,30,31,32,33,34]. Previous studies have shown that b-FGF has several positive effects on bone formation [36,37,38]. Acidic gelatin (AG), produced by collagen degradation, maintains electrostatic interaction and slowly releases b-FGF [39,40]. AG impregnated with b-FGF solution accelerates bone formation in critical-sized defects in animal models [41,42,43].

Thus far, four materials and growth factors such as Col, Hap, AG, and b-FGF were referred to form bone substitute materials. The composites comprised of two to three selected materials and growth factor have recently been utilized for regenerative biomaterials, as follows. Du et al. [44] used acellular dermal matrix (mostly, Col) loading with b-FGF to accelerate bone regeneration by recruitment, proliferation, and sustained osteo-differentiation of mesenchymal stem cells. Ueno et al. [45] employed Col sheet loaded with b-FGF fused to the Col-binding domain with bone allogenic graft to increase callus volume in the femoral bone defect model. Madani et al. [46] evaluated quercetin-loaded gelatin/tragacanth/nano-Hap composite in the cell culture test as a bone tissue engineering scaffold. Yamaguchi et al. [47] studied b-FGF-containing Hap/Col in prefabricated vascularized allo-bone grafts for bone union enhancement. Santhakumar et al. [48] facilitated in situ precipitation of amorphous calcium phosphate (analogue of Hap) nanoparticles within three-dimensional porous Col sponges and immobilized b-FGF for bone tissue engineering. Matsumine et al. [49] examined b-FGF-impregnated Col-gelatin sponge for full-thickness skin reconstruction and obtained favorable results. Nakamura et al. [50] reported that bone regeneration was accelerated in rat horizontal alveolar bone defect model using Col-binding b-FGF combined with Col scaffolds. Imada et al. [51] found that the use of bFGF-containing gelatin hydrogel prevented tooth extraction-triggered bisphosphonate-related osteonecrosis of the jaws. Sohn et al. [52] succeeded in maintaining edentulous alveolar ridge preservation using b-FGF in combination with collagenated biphasic calcium phosphate. Kobayashi et al. [53] confirmed that alpha-tricalcium phosphate with immobilized b-FGF enhanced bone regeneration in a canine mandibular bone defect model.

The combination of quaternary compositions such as Col, Hap, AG, and b-FGF has, however, not been reported yet. Its combination might produce very effective bone substitute materials in the next generation. It has the potential to up-grade bone-forming capability, the level of which could be arbitrarily adjusted by the mixing ratios of the compositions, the cross-link levels of Col and AG, and amounts of b-FGF loaded. This study was a first step to realize this objective. We believed that this pioneering research has novelty and value in academic fields.

The Col membrane was very important in assisting in forming bones by Col/Hap composite granules and was self-prepared in this study. Its usage, preparation method, and characterization are mentioned in Appendix A: Self-Preparation of Collagen (Col) Membranes (Appendix A) [49,54,55,56,57,58,59,60,61,62,63,64,65,66,67,68,69,70,71,72,73]. The formed Col membranes were employed in animal studies, which are explained later. These characterization studies included different Col base material produced for Col/Hap composites. Self-preparation of the Col membrane was the second-priority purpose of this investigation.

Upon consideration of all above information, the purposes of this investigation were (1) to prepare Col/Hap composites by freeze drying and alternate immersion methods and characterize materialistically the composites; (2) to infiltrate acidic gelatin into the composites, followed by freeze drying and b-FGF impregnation; and (3) to implant the wet constructs (collagen/hydroxyapatite/acidic gelatin/b-FGF, coded by Col/Hap/AG/b-FGF) in critical-size defects of rats, the top of which was covered by the prepared membrane, followed by soft X-ray measurements and histological observations so that the constructs could be evaluated as new, novel osteo-conductive bone substitute materials.

## 2. Materials and Methods

### 2.1. Materials

#### 2.1.1. Col/Hap Composite

We used virus-free, medical-grade Col pellets (NMP collagen PS; Nippon Meat Packers Inc., Tokyo, Japan) extracted from porcine skin using pepsin. The pellets mainly comprised type I collagen and small quantities of type III collagen. We used calcium chloride (CaCl_2_; Kanto Chemical Co., Tokyo, Japan), sodium dihydrogen phosphate dihydrate (NaH_2_PO_4_·2H_2_O; Junsei Chemical Co., Tokyo, Japan), Tris hydrochloride (Tris HCl; Gibco BRL, ThermoFisher Scientific, Waltham, MA, USA), 1 N sodium hydroxide (NaOH) solution (Kanto Chemical Co., Tokyo, Japan), and 1N hydrochloric acid (HCl) solution (Nacalai Tesque Co., Kyoto, Japan).

#### 2.1.2. Col/Hap Composite Complexed with AG and b-FGF

AG was obtained from the denatured product of alkaline-treated bovine bone (G-2700P; Nitta Gelatin Co., Osaka, Japan). Water-soluble ethylene glycol diglycidyl ether (EGDE) (Denacol EX-810; Nagase Chemtex, Osaka, Japan) was used for chemical cross-linking of AG. For b-FGF, Fibroblast Spray 500 (500 μg/5 mL; Kaken Pharmaceutical Co., Tokyo, Japan) was used.

### 2.2. Preparation of Biomaterials

#### 2.2.1. Preparation of Col/Hap Composite Granules

Col pellets (1 g) were dissolved in 28 mL of distilled water in a 50 mL polystyrene conical tube at 4 °C. The acidic solution was neutralized using 0.1 N NaOH solution (6.5 mL) in three rectangular plastic plates (84 × 54 × 12 mm^3^) to achieve a Col gel pH of 7.5. The Col gel was frozen at −80 °C for 12 h and freeze dried for 12 h. The resultant sponge was cross-linked using DHT treatment at 140 °C for 24 h in a vacuum dry oven.

The Col sponge was processed using an alternate immersion method (Figure 1). In brief, three sponge sheets produced from 1 g of pellets were cut into 0.5–1 mm granules using scissors. The granules were packed in nine Nylon meshes (Mesh Pack C, 60 mm × 80 mm; Sansho Co., Tokyo, Japan). The granules in the mesh were immersed in 100 mL of Tris-HCl buffered solution containing 200 mM CaCl_2_ (pH = 7.4) for 20 min at 37 °C, blot dried using a paper cloth, immersed in 100 mL of Tris-HCl buffered solution containing 120 mM NaH_2_PO_4_ (pH = 9.3) for 20 min at 37 °C, and blot dried using a paper cloth to complete one cycle of the alternate immersion method. The immersion cycle was repeated five times (AI 20 min 5Cy Col/Hap). The pH was adjusted with NaOH and HCl solutions using a pH/ion meter (F-24; Horiba Ltd., Kyoto, Japan). The composite sponges were produced by altering the immersion time of five cycle repetitions to 60 min (AI 60 min 5Cy Col/Hap). Two types of composite sponges were dried in a vacuum dry oven (VO-300; AS One) at 56 °C for 48 h. Control collagen containing only granules without the use of alternate immersion was also prepared (Col control).

#### 2.2.2. Complex of Col/Hap Composite Granules with AG and b-FGF Loading

The preparation of AG is briefly described below. First, AG (4 g) was dissolved in distilled water (36 mL) at 37 °C. The gelatin solution (800 μL) was poured into the 1.5 mL Eppendorf tube, followed by mixing with the cross-linking solution (200 μL). The cross-linking solution consisted of distilled water (95 mL), EGDE cross-linker (5 mL), and NaCl (11.7 g), with the pH maintained at 7. Chemical cross-linking was performed by storing the solution at 4 °C for 3 days. The product from the 18 tubes was purified using dialysis (Mw  =  12,000–14,000, Code 3-25; Thermo Fisher Scientific, Waltham, MA, USA) against exchanged distilled water (1 L), three times for 3 days. The volume of the cross-linked AG was increased by 2.5-fold relative to that of the original non-cross-linked gelatin.

Two collagen/apatite composite granules (AI 20 min 5Cy Col/Hap and AI 60 min 5Cy Col/Hap) and Col control granules were infiltrated by ample cross-linked AG, followed by freezing at −80 °C for 12 h and freeze drying for 24 h. The three types of dried gelatin infiltrated granules (Col control + AG, AI 20 min 5Cy Col/Hap + AG and AI 60 min 5Cy Col/Hap + AG) were sterilized using ethylene oxide gas and stored in a desiccator.

Immediately prior to the animal experiments, the three types of dried gelatin-infiltrated granules were immersed in ample b-FGF solution for 60 min at 4 °C to form three wet constructs (Col control + AG + b-FGF, AI 20 min Co/Hap + AG + b-FGF, and AI 60 min Co/Hap + AG + b-FGF).

#### 2.2.3. Sample Codes

Table 1 shows the details of code, composition, and preparation process (major part) of samples examined so that inter-relationships among samples could be better understood.

### 2.3. Characterization of Biomaterials

#### 2.3.1. SEM/Energy-Dispersive Spectroscopy (EDS) and Scanning Electron Microscopy (SEM) Analyses

The morphological and chemical properties of the outer surfaces of the two Col/Hap composite granule samples (AI 20 min 5Cy Col/Hap and AI 60 min 5Cy Col/Hap) were examined using SEM (SU8010; Hitachi High-Tech Co., Tokyo, Japan) and EDS (JSM-7100F; Joel Co., Tokyo, Japan) (*n* = 1) at an accelerating voltage of 10 kV.

The outer and cross-sectional surfaces of three types of dried gelatin-infiltrated granules (Col control + AG, AI 20 min 5Cy Col/Hap + AG and AI 60 min 5Cy Col/Hap + AG) were examined (*n* = 1 for both) using SEM (SU8010; Hitachi High-Tech Corp., Tokyo, Japan) at an accelerating voltage of 15 kV after plasma coating with OsO_4_.

#### 2.3.2. X-ray Diffraction (XRD) Analysis

The crystallographic states of Col and Col/Hap composite granules (Col control, AI 20 min 5Cy Col/Hap, and AI 60 min 5Cy Col/Hap) were examined (*n* = 1 for all) using XRD (D8 Discover; Bruker AXS, Billerica, MA, USA), CuKα radiation, and an accelerating voltage of 40 kV. Pure Hap produced by high-temperature sintering (187-37; Taihei Chemical Industry Co., Osaka, Japan) was used as standard for comparison.

#### 2.3.3. Fourier-Transform Infrared Spectroscopy (FTIR)

The organic functional groups in Col and Col/Hap composite granules (Col control, AI 20 min 5Cy Col/Hap, and AI 60 min 5Cy Col/Hap) were examined (*n* = 1 for all) using FTIR equipped with attenuated total reflectance attachment (Nicolet6700; Thermo Fisher Scientific, Waltham, MA, USA). Hap standard was also used for comparison.

#### 2.3.4. Growth Factor b-FGF Levels in the AG-Infiltrated and b-FGF-Loaded Granules

The rate of absorption (wt%) of b-FGF solution by AG-infiltrated granules (Col control + AG, AI 20 min 5Cy Col/Hap + AG, and AI 60 min 5Cy Col/Hap + AG) was calculated (*n* = 6 for all) by measuring the weight of the granules before and after immersion of AG. The dry and wet granules were compacted into a stainless steel hole mold (6 mm in diameter and 1 mm in height), with a size equivalent to the defect size created on the cranial bones of rats, and analyzed in terms of the bulk density and b-FGF absorption rate, respectively. The absorption rate (%) was calculated by the weight of absorbed b-FGF solution (g) divided by the weight of the original granule before dipping (g) and multiplying by 100. The quantity of b-FGF in three implanted granules with b-FGF (Col control + AG + b-FGF, AI 20 min Co/Hap + AG + b-FGF, and AI 60 min Co/Hap + AG + b-FGF) was estimated.

### 2.4. Animal Experiments

#### 2.4.1. Operation

We used 18 male Wistar rats weighing 340 ± 16 g. The rats were housed in separate cages (three rats per cage) and provided with standard diet and water ad libitum. Using anesthesia with a mixture of isoflurane (3% vol) and oxygen (0.5 L/min) gas generated by a carburetor (IV-ANE; Olympus, Tokyo, Japan), the centers of the rat calvariae were shaved, sterilized with 10% povidone iodine, and injected with a local anesthetic (0.2 mL, 2% lidocaine with 1:80,000 epinephrine). Then, full-thickness periosteum flaps were elevated, and bone defects were created using a trephine bur (6 mm diameter; Implant Re Drill System, GC, Tokyo, Japan). Six specimens from each granule sample (Col control + AG + b-FGF, AI 20 min 5Cy Col/Hap + AG + b-FGF, and AI 60 min 5Cy Col/Hap + AG + b-FGF) were implanted in the calvarial bone defects (Figure 2), whereas six holes were left empty (defect only). The defects were covered with the self-prepared Col membrane. The flaps were repositioned and sutured using soft nylon (Softretch 4-0, GC). At 8 weeks after surgery, the rats were sacrificed using CO_2_ inhalation. The animal experiments were performed in accordance with the Guidelines for the Care and Use of Laboratory Animals and approved by the Institutional Ethics Committee of Iwate Medical University (19 March 2021; approval no.: #02-035).

#### 2.4.2. Soft X-ray Measurements

We performed soft X-ray (M60; Softex, Tokyo, Japan) to evaluate new bone formation at the cranial critical defects after implantation of the three granule types with b-FGF (Col control + AG + b-FGF, AI 20 min Co/Hap + AG + b-FGF, and AI 60 min Co/Hap + AG + b-FGF) as well as at untreated defects. Figure 3 illustrates the top and cross-sectional views of the bone formation process at the rat cranial bone defects. The original bone defect is indicated by solid lines, and newly formed bone is indicated by broken lines. Small island-like bone fragments inside and around the bone defect as well as newly extended bone are indicated by green color. Soft X-ray was performed to identify new bone formation based on the X-ray grey values of the areas that corresponded to the original defect. The images were analyzed using ImageJ software (1.53k; National Institutes of Health, Bethesda, MD, USA).

#### 2.4.3. Histological Analysis

##### Decalcified Tissue Samples

After feeding for 8 weeks, cranial bones with and without granule samples were removed using a diamond saw (MC-201 Microcutter; Maruto, Tokyo, Japan) or scissors and fixed in 10% neutral buffered formaldehyde equivalent (Mildform; Wako Chemicals, Osaka, Japan) for 4 weeks at 4 °C. Then, the bones were decalcified in 0.5 wt% ethylene diamine tetra-acetate solution (Decalcifying solution B; Wako Chemicals, Osaka, Japan) for 4 weeks at 4 °C. Next, the bones were treated with graded alcohol and xylene and embedded in wax. The specimens in the wax were cut into 5 μm sections using a microtome (IVS-410; Sakura Finetek, Tokyo, Japan). The sections were stained with hematoxylin and eosin (HE) and observed using fluorescence microscopy (All-in-one BZ-9000; Keyence, Osaka, Japan).

##### Non-Decalcified Tissue Samples

Fluorescent double staining was performed on two of the six rats implanted with the granules. Sequential labeling was performed to evaluate postoperative bone formation and remodeling. Rats received an intraperitoneal injection of tetracycline (TC; oxytetracycline hydrochloride; Nacalai Tesque Co., Kyoto, Japan) (3 mg/100 g body weight) dissolved in physiological saline solution (Otsuka Pharmaceutic Co., Tokyo, Japan); 0.4 mL) at 5 and 6 weeks after the surgery, followed by injection of calcein (CL; Wako Chemicals, Osaka Japan; 1 mg/100 g body weight) in 0.4 mL of physiological saline solution at 7 weeks and 7 weeks + 5 days (2 days before sacrifice) after the surgery. The rat calvariae embedded with granules were processed to observe the non-decalcified histological appearance. After fixation in 70% ethanol (99.5% pure, Junsei Chemical Co., Tokyo, Japan) at 4 °C for 1 week, the samples were dehydrated in a graded series of ethanol (1 day at each concentration) and placed in pure acetone (Kanto Chemical Co., Tokyo, Japan) for 24 h. The samples were then stained with Villanueva solution (222-01445; Wako Chemicals, Osaka, Japan), embedded in methylmethacrylate for 4 days, and chemically polymerized for 10 days. The non-decalcified resin blocks (almost 15 mm × 15 mm × 20 mm) were cut in the sagittal plane using a circular diamond cutter (MC-201 Microcutter; Maruto, Tokyo, Japan). The sections were attached to plastic slides, ground to a thickness of 20 μm using a precision lapping machine (ML-110N; Maruto, Tokyo, Japan), and manually polished. Histological analysis of the sections was performed using fluorescence microscopy (All-in-one BZ-9000; Keyence, Osaka, Japan)) for Villanueva-stained images and single CL-fluor-labeling, with confocal laser scan microscopy (C1si; Nikon Co., Tokyo, Japan) used for dual TC&CL-fluor-labeling analysis.

### 2.5. Statistical Analyses

Free statistical software (EZR version 1.55, Saitama Medical Center, Jichi Medical University, Saitama, Japan) [74] was used for nonparametric tests, such as Kruskal–Wallis test. The null hypothesis was rejected at *p* < 0.05.

## 3. Results

### 3.1. Col/Hap Composites by Alternate Immersion Method

#### 3.1.1. SEM/EDS

Figure 4a shows a highly magnified SEM image of a Col/Hap composite (AI 20 min 5Cy Col/Hap). The composite surface was covered by small, flaky crystals. The short and long dimensions of the crystals were 295 and 472 nm (*n* = 10 each), respectively. Figure 4b indicates the chemical composition based on EDS point analysis of “x1” in Figure 4a. The crystals contained calcium, phosphorus, oxygen, and carbon. Figure 4c shows a highly magnified SEM image of another collagen/apatite composite (AI 60 min 5Cy Col/Hap). The composite surface was also covered by small, flaky crystals. The short and long dimensions of the crystals of AI 60 min 5Cy Col/Hap composite were 352 and 903 nm (*n* = 10 each), respectively, which were larger than those of the former crystals on AI 20 min 5Cy Col/Hap composite. Figure 4d indicates the EDS spectrum obtained using compositional analyses of “x2” in Figure 4c. The crystals were composed of calcium, phosphorus, oxygen, and carbon. Both crystals were small (nano-sized).

#### 3.1.2. XRD

Figure 5 shows the XRD patterns of a Col/Hap composite (AI 20 min 5Cy Col/Hap), Col/Hap composite (AI 60 min 5Cy Col/Hap), Hap standard, and Col control. Hap standard had sharp and distinct characteristic peaks of highly crystalline Hap [75], whereas Col control had no specific peak. However, the two composites contained low-crystalline Hap with dull and ambiguous hydroxyapatite peaks [76].

#### 3.1.3. FTIR

Figure 6 presents the FTIR charts of a Col/Hap composite (AI 20 min 5Cy Col/Hap), Col/Hap composite (AI 60 min 5Cy Col/Hap), Hap standard, and Col control. Hap standard had PO_4_ peaks [75], whereas Col control had amide (-CO-NH_2_-) I and II peaks. Two composites had PO_4_ and amide peaks, attributable to Hap and Col, respectively.

### 3.2. Complexes with AG and b-FGF Loading

#### 3.2.1. SEM

Figure 7a–c show SEM photomicrographs of the outermost and cross-sectional surfaces of dry gelatin-infiltrated Col control powder (Col control + AG) and two alternate immersed composite granules (AI 20 min 5Cy Col/Hap + AG and AI 60 min 5Cy Col/Hap + AG), respectively. The AG overlapped the surface of Col and Col/Hap composites, covering the precipitated crystals. However, some pores were present cross-sectionally on the three granules despite AG filling.

#### 3.2.2. Quantity of b-FGF Loading to Col/Hap/AG Granules

Figure 8 top-left in blue color, top-right in orange color, and bottom-right in green color present the relative density (%) (bulk density (mg/mm^3^)), absorption rates of the b-FGF solution (%), and estimated b-FGF quantity in the stainless-steel die, respectively, of the three AG-infiltrated granules ((a) Col control + AG, (b) AI 20 min 5Cy Col/Hap + AG, and (c) AI 60 min 5Cy Col/Hap +AG). Two alternate immersed composite granules (AI 20 min 5Cy Col/Hap + AG and AI 60 min 5Cy Col/Hap + AG) had a similar density, with approximately 20% larger granules than those of collagen control (Col control + AG) (Figure 8 top-left). After dipping of b-FGF solution, the control sample (Col control + AG) exhibited the highest absorption capacity for the b-FGF solution (%), followed by AI 60 min 5Cy Col/Hap +AG granules; AI 20 min 5Cy Col/Hap + AG had the lowest absorption capacity (Figure 8 top-right). The b-FGF solution was water-based, while b-FGF itself was crystalline protein powder. Considering that the concentration of b-FGF solution was 100 μg/1 mL, the b-FGF quantity could be estimated by absorbed b-FGF solution. The b-FGF quantities absorbed by the three granule samples (Col control + AG, AI 20 min Co/Hap + AG, and AI 60 min Co/Hap + AG) in the metal die were calculated to be 3.5–5 μg (Figure 8 bottom-left).

### 3.3. Animal Studies of Col/Hap/AG/b-FGF Constructs

#### 3.3.1. Soft X-ray Analyses

Figure 9a–d show representative soft X-ray images of rat cranial bone defects with and without the implanted b-FGF immersed granules ((a) defects only, (b) Col control + AG + b-FGF, (c) AI 20 min Co/Hap + AG + b-FGF, and (d) AI 60 min Co/Hap + AG + b-FGF). Figure 10 shows the X-ray grey values of rat cranial bone defects with and without b-FGF-loaded granules (*n* = 6 for each granule type). Weak bone formation occurred at the rat cranial bone defects. The three implanted granules increased the X-ray grey values. The grey values were significantly higher at the cranial bone defects with AG-filled composite (AI 20 min 5Cy Col/Hap + AG + b-FGF) compared to the unfilled defects (Col control + AG + b-FGF) (*p* < 0.05).

#### 3.3.2. Decalcified Tissue Histology

Figure 11a–d show HE-stained histological images of rat cranial bone defects with and without implanted b-FGF-loaded granules, including (a) those with defect only, (b) Col control + AG + b-FGF, (c) AI 20 min 5Cy Col/Hap + AG + b-FGF, and (d) AI 60 min 5Cy Col/Hap + AG + b-FGF. The defects showed varying degrees of bone extension from the defect edges and sporadic island-like small bone formation inside the defect.

Figure 12a shows magnified images of the dotted square zone (“X1”) in Figure 11c. Figure 12b is a magnified image of the bone head shown in Figure 12a, which revealed extensive bone extension. The entire bone shown in Figure 12a was the newly formed extended bone. The extended bone consisted of an array of discrete, newly formed island-like bones. Figure 13a is a magnified image of the dotted square zone (“X2”) in Figure 11d. Figure 13b is a magnified image of Figure 13a, showing the bone head. An island-like bone was formed inside the defect and in front of the right side of the pre-existing bone.

#### 3.3.3. Non-Decalcified Tissue Histology

Figure 14a shows a Villanueva-stained image of a rat cranial bone defect filled with AG-infiltrated and b-FGF-loaded control collagen (Col control + AG + b-FGF). There was extensive bone formation on the left side of the preexisting bone in the bone defect. Figure 14b is a magnified CL-fluor-labeled image corresponding to “X1” in Figure 14a. Figure 14c is a magnified CL-fluor-labeled image of “X2” in Figure 14b. In the final week, bone formation was noted in widespread areas, appearing as thickened bone on the inner and outer sides.

Figure 15a shows a Villanueva-stained image of a rat cranial bone defect filled with an AG-infiltrated and b-FGF-loaded Col/Hap composite (AI 20 min 5Cy Col/Hap + AG + b-FGF). There was relatively slow bone extension from both bone edges, but small particles were found within the bone defect. Figure 15b,c show the magnified and CL-fluor-labeled images of “X1” in Figure 15a, respectively. In the final week, slight bone extension was observed. Figure 15d,e show the magnified and TC&CL-fluor-labeled images of “X2” in Figure 15a, respectively. The particles were CL-stained, which revealed bio-absorption of materials and an initial ossification reaction (i.e., osteoid formation) [77]. Figure 15f,g show magnified and TC&CL-fluor-labeled images of “X3” in Figure 15a, respectively. In the final week, slight bone extension was observed on the right side of pre-existing bone. Dual fluor-labeling revealed bone formation over the final 3 weeks.

Figure 16a shows a Villanueva-stained image of a rat cranial bone defect filled with another AG-infiltrated and b-FGF-loaded collagen/apatite composite (AI 60 min 5Cy Col/Hap + AG + b-FGF). There was prominent bone extension from the left side of the bone edge (exceeding 2.5 mm), while slight bone extension was also noted from the right side. Figure 16b,c show magnified and CL-fluor-labeled images of “X1” in Figure 16a, respectively. In the final week, rapid bone formation was observed. Figure 16d is a magnified TC&CL-fluor-labeled image of “X3” in Figure 16c. Bone formation became prominent in the final 3 weeks. Figure 16e,f show magnified and CL-fluor-labeled images of “X2” in Figure 16a, respectively. In the final week, sluggish bone formation was observed. Figure 16g shows a magnified TC&CL-fluor-labeled image of “X4” in Figure 16f. The bone edge slowly extended outward in the final 3 weeks.

## 4. Discussion

Figure 17 illustrates the development of construct granules (Col/Hap/AG/b-FGF) and the relationship of constituent materials. Table 2 describes the roles of materials contained in the construct. Col is a space-creating scaffold [78,79]. Low-crystalline Hap is bio-absorbable and osteo-conductive [80,81]. AG is both a scaffold and excellent b-FGF carrier [82,83]. The growth factor b-FGF has several therapeutic effects, such as promoting angiogenesis and recruiting stem cells Appendix A: Tissue Engineering and Therapeutic Effects of b-FGF (Appendix A) [28,29,30,31,32,33,34]) [39].

We successfully coated low-crystalline Hap on a Col sponge using an alternate immersion method (Figure 4, Figure 5 and Figure 6). Although several methods have been used for Hap coating of Col, including electro-phoretic deposition [84] and biomimetic methods [85], we selected an alternate immersion method [86,87,88,89,90] because of its simplicity and low cost. A possible drawback of the immersion method was the weak bond between small Hap crystals and Col surfaces. This drawback may be overcome by gelatin filling. The alternate immersion method is associated with the bio-mimetic process at body temperature. The quality of newly formed Hap is also important. Low-crystalline Hap is bioabsorbable and can be replaced by new bone [91,92], whereas highly-crystalline Hap is almost non-bio-absorbable, hindering the formation of new bone [91].

There are certain important considerations with respect to infiltration of AG into Col/Hap composite granules. AG was cross-linked using a safe cross-linker (EGDE) under mild conditions. Cross-linking was necessary to prolong the in vivo life of gelatin to 2–3 weeks [93]. Otherwise, the gelatin would be eliminated within 3 days [94]. SEM analysis confirmed that AG completely covered the inner surfaces of collagen and collagen/apatite composite granules (Col control + AG, AI 20 min 5Cy Col/Hap +AG, and AI 60 min 5Cy Col/Hap + AG) with a porous inner structure (Figure 7). The quantity of b-FGF absorbed by the three granules in the metal die, equivalent to the volume of a cranial bone defect, was almost 3.5–5 μg (Figure 8), which was lower than the doses used in previous bone formation studies (e.g., 115 μg in a large defect) [34].

The rat cranial bone defect model is commonly used to evaluate the bone-forming capability of biomaterials [95,96,97,98,99]. We successfully produced new bone in rat cranial bone defects over 8 weeks; the new bone covered 20–35% of the cranial defect after implantation of AG-infiltrated and b-FGF-loaded Col/Hap composite granules (AI 20 min 5Cy Col/Hap + AG + b-FGF and AI 60 min 5Cy Col/Hap + AG + b-FGF) (Figure 9). This level of bone regeneration was moderately successful compared to previous bone formation studies [100,101]. There was significant variation in soft X-ray grey values, reflecting differences in bone regeneration between defects with and without AG-infiltrated and b-FGF-loaded granules. Figure 10 illustrates the combination associated with statistically significant bone regeneration.

We did not effectively use periosteum for bone formation [102]. Intraoperatively, we created a full flap and completely removed the cranial bone periosteum. After repositioning, the periosteum did not completely cover the bone defect. If the periosteum had completely covered the cranial bone defect, bone formation would have been considerably enhanced [103]. Improvement in operator skill is also important to enhance bone formation.

The quality and nature of the bone formed in the rat cranial bone defects are important considerations. Figure 3 illustrates the method of bone formation in rat cranial bones (Figure 11, Figure 12, Figure 13, Figure 14, Figure 15 and Figure 16). New bone was formed by extension from preexisting bone at the defect edge and from small island-like bones. Intramembranous ossification may explain the new bone formation, particularly of small island-like bones [104,105]. The extended bones often developed via the aggregation of small, arrayed island-like bones (Figure 12a and Figure 16a). The island-like small bones may have merged and united with the preexisting bone to achieve complete healing. We also found small remaining particles (AI 20 min 5Cy Col/Hap +AG + b-FGF) within the cranial bone defect (Figure 15 a,d,e). The particle was CL-fluor-stained [106,107], indicating that the liberated calcium ions were bound to CL; this may have been due to particle dissolution, replacement of particles with the newly formed bone precursor (i.e., osteoid), or both [108].

When used as a bone substitute, the implant material may last for longer than 2–3 months while slowly releasing growth factors [109]. The addition of osteoconductive materials to implant material may improve bone formation. The b-FGF-loaded granules prepared in this study (Col control + AG + b-FGF, AI 20 min 5Cy Col/Hap + AG + b-FGF, and AI 60 min 5Cy Col/Hap + AG + b-FGF) disappeared earlier than in the afore-mentioned period, leaving behind few granules. The in vivo longevity of b-FGF-impregnated granules and the degree of bone formation is enhanced by increased cross-linking of collagen and AG [110,111].

## 5. Conclusions

We prepared Col/Hap composite granules using the alternate immersion method and infiltrated them with cross-linked AG, followed by freeze drying. The growth factor b-FGF was impregnated into the prepared granules, which produced Col/Hap/AG/b-FGF constructs. The wet quaternary granules were implanted into rat cranial bone defects and evaluated using soft X-ray measurements and histological analysis. We self-prepared collagen membranes using chemical cross-linking to cover bone defects with and without filled constructs.

Despite its limitations, we can draw several important conclusions from our study of Col/Hap/AG/b-FGF granules. First, the self-prepared Col membrane provided a protective barrier and covering material for the defects. Second, instrumental analyses (SEM, SEM/EDS, XRD, and FTIR) showed that alternate immersion promoted small low-crystalline Hap precipitation on the collagen matrix. Third, the placement of the Col/Hap/AG/b-FGF construct increased new bone formation in the cranial bone defect compared to the defect without placement. Fourth, the newly formed bones extended from the bone defect edge and produced small island-like bones. Finally, the prepared construct (Col/Hap/AG/b-FGF) produced a moderate amount of new bone. Further materialistic studies are required to develop methods to improve bone formation.

## Figures and Tables

**Figure 1 materials-15-08802-f001:**
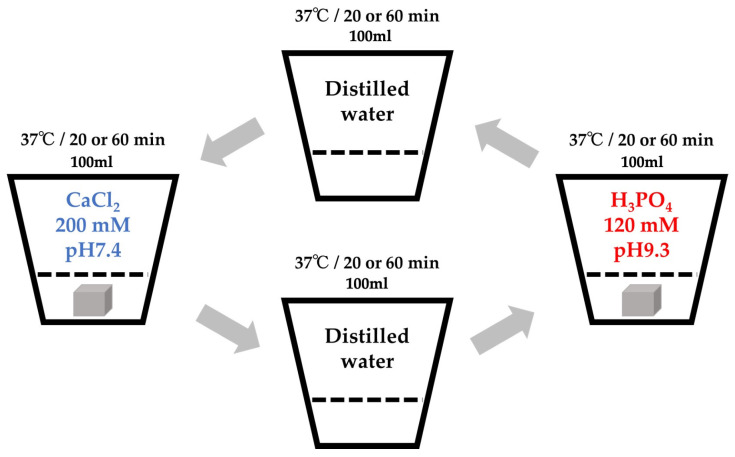
Schematic of the alternate immersion method.

**Figure 2 materials-15-08802-f002:**
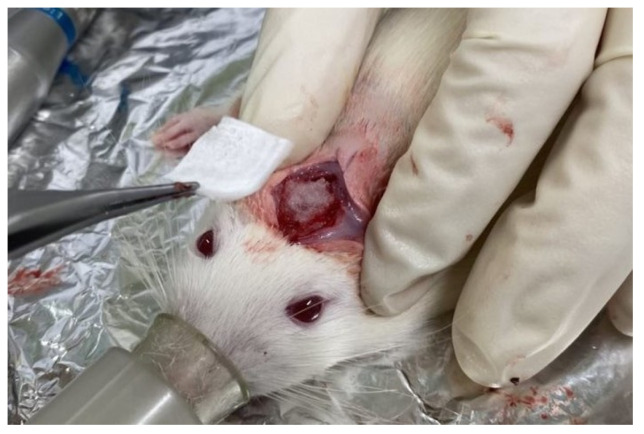
Photograph of the prepared construct granules in a rat cranial bone defect and the covering collagenous membrane.

**Figure 3 materials-15-08802-f003:**
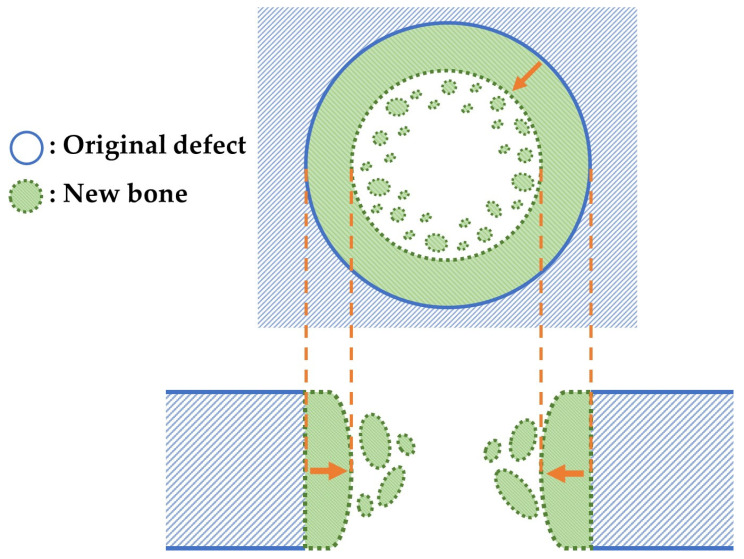
Top and cross-sectional views of new bone formation in a cranial bone defect. Note: the original defect area is denoted by solid lines. The extended bone area is denoted by broken lines and green color.

**Figure 4 materials-15-08802-f004:**
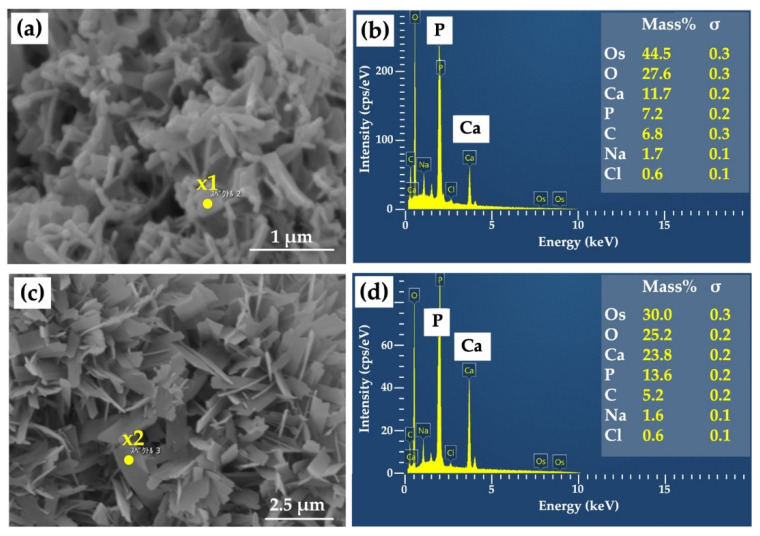
(**a**) High-magnification SEM image of a collagen/apatite composite (AI 20 min 5Cy Col/Hap); (**b**) EDS spectrum and compositional analyses of “x1” in Figure 4a; (**c**) high-magnification SEM image of another collagen/apatite composite (AI 60 min 5Cy Col/Hap); (**d**) EDS spectrum and compositional analyses of “x2” in Figure 4c. Note: (**a**,**c**) 10,000× magnification.

**Figure 5 materials-15-08802-f005:**
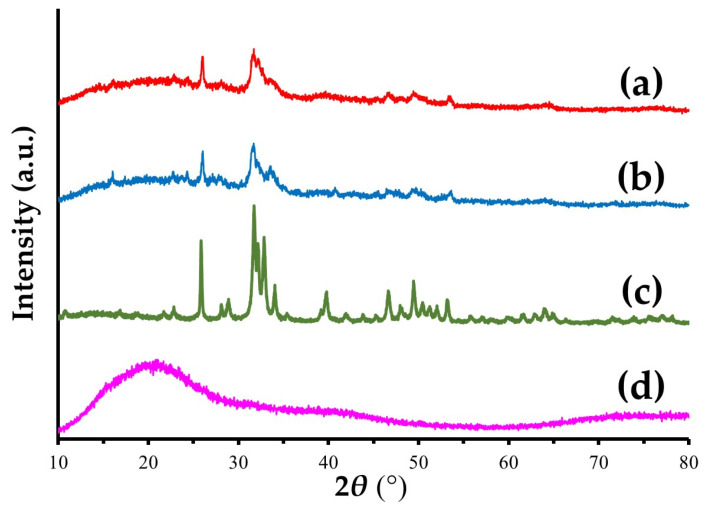
XRD patterns of (**a**,**b**) two Col/Hap composites (AI 20 min 5Cy Col/Hap and AI 60 min 5Cy Col/Hap), (**c**) Hap standard, and (**d**) Col control.

**Figure 6 materials-15-08802-f006:**
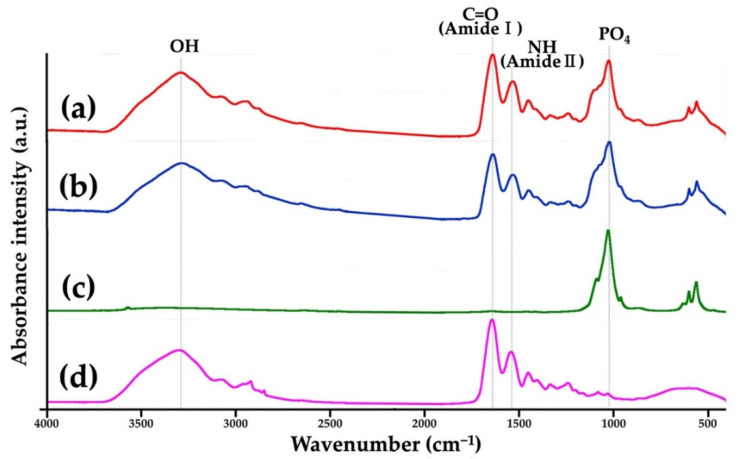
FTIR charts of (**a**,**b**) two Col/Hap composites (AI 20 min 5Cy Col/Hap and AI 60 min 5Cy Col/Hap), (**c**) Hap standard, and (**d**) Col control.

**Figure 7 materials-15-08802-f007:**
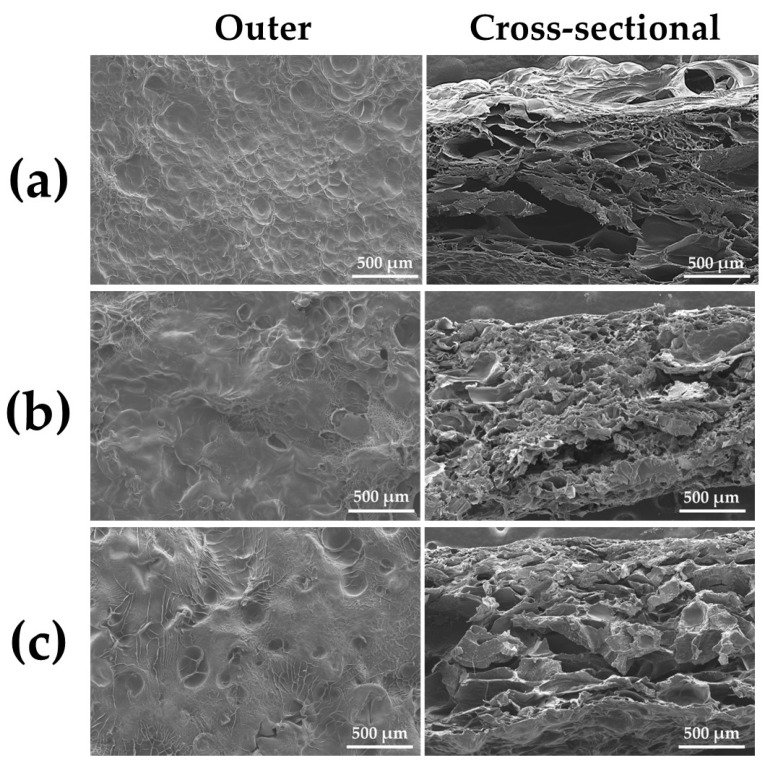
SEM photomicrograph of the outermost (50× magnification) (**left**) and cross-sectional (50× magnification) (**right**) surfaces of (**a**) dry gelatin-infiltrated control Col powder (Col control + AG) and two alternate immersed composite granules: (**b**) (AI 20 min 5Cy Col/Hap + AG) and (**c**) (AI 60 min 5Cy Col/Hap + AG).

**Figure 8 materials-15-08802-f008:**
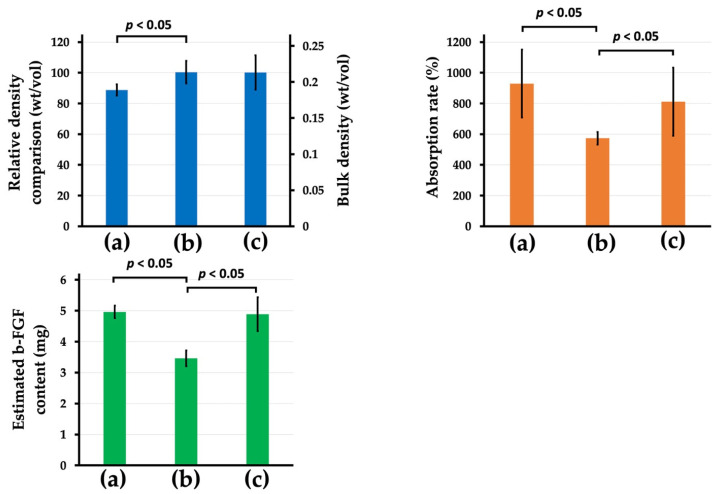
(**Top-left**) Relative density (%) (bulk density (mg/mm^3^)), (**top-right**) the b-FGF solution absorption rates (%), and (**bottom-left**) estimated b-FGF quantity in the stainless steel die of the three AG-infiltrated granules: (a) Col control + AG, (b) AI 20 min 5Cy Col/Hap + AG, and (c) AI 60 min 5Cy Col/Hap + AG. (*n* = 6 for each) Note. The concentration of b-FGF solution was 100 μg/1 mL.

**Figure 9 materials-15-08802-f009:**
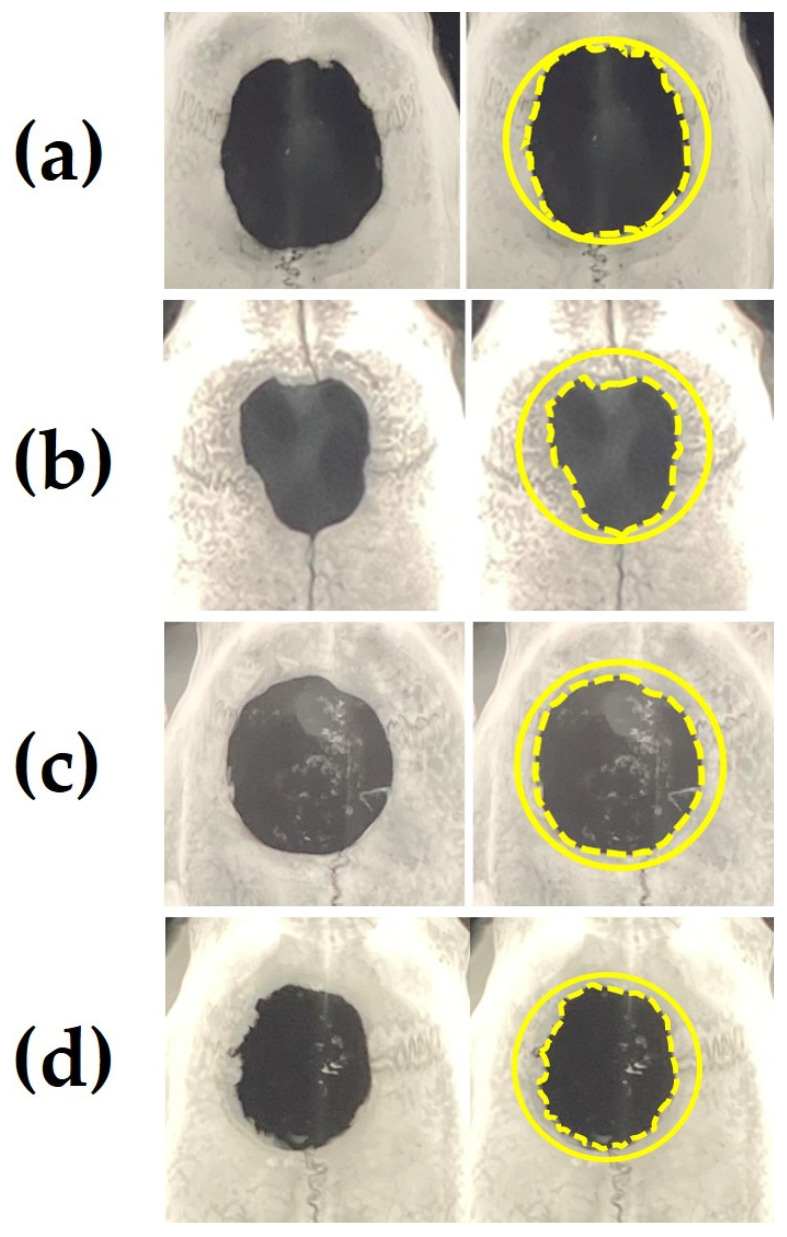
Representative soft X-ray images of a rat cranial bone defect with and without three AG-infiltrated and b-FGF-loaded granules, including (**a**) defect without granules and defects with (**b**) Col control + AG + b-FGF, (**c**) AI 20 min 5Cy Col/Hap + AG + b-FGF, and (**d**) AI 60 min 5Cy Col/Hap +AG + b-FGF. Note: left, raw data; right, the original defect and edges of extending bones are depicted by solid circles and broken lines, respectively.

**Figure 10 materials-15-08802-f010:**
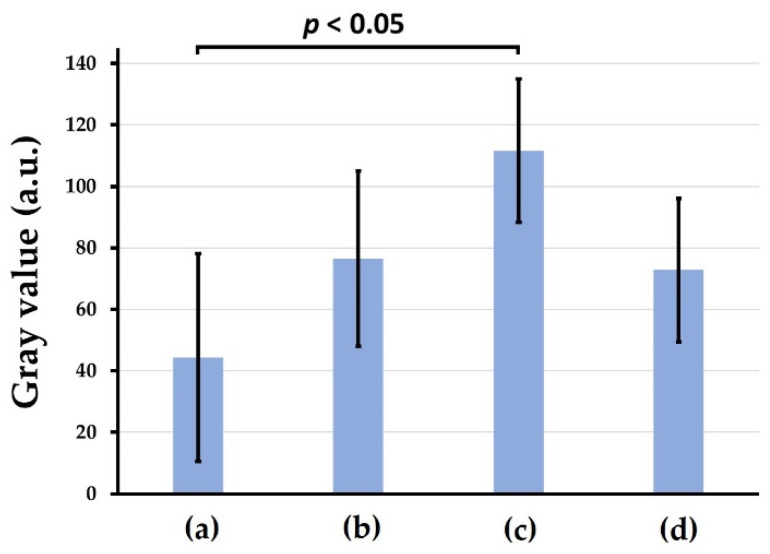
X-ray grey values of rat cranial bone defects with and without the three b-FGF-loaded granules: (**a**) defect only, (**b**) Col control + AG + b-FGF, (**c**) AI 20 min 5Cy Col/Hap + AG + b-FGF, and (**d**) AI 60 min 5Cy Col/Hap + AG + b-FGF. (*n* = 6 for each).

**Figure 11 materials-15-08802-f011:**
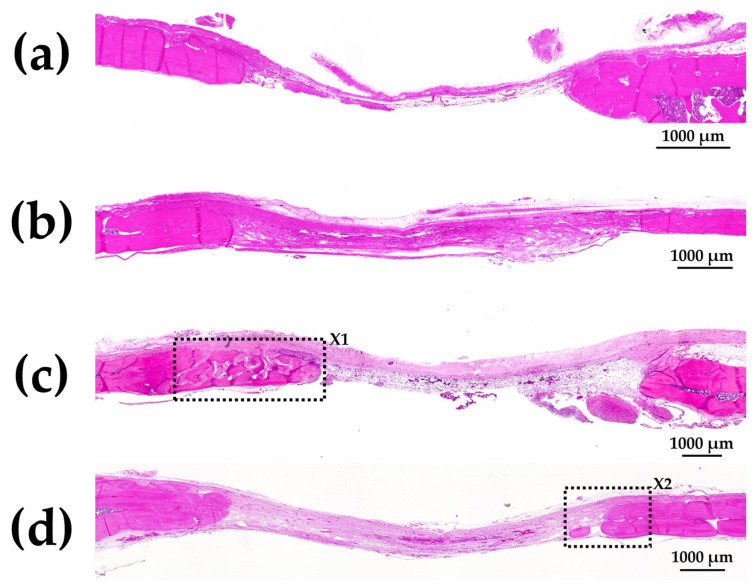
HE-stained histological images of rat cranial bone defects with and without the three granules: (**a**) defect only, (**b**) Col control + AG + b-FGF, (**c**) AI 20 min 5Cy Col/Hap + AG + b-FGF, and (**d**) AI 60 min 5Cy Col/Hap + AG + b-FGF. (4× magnification).

**Figure 12 materials-15-08802-f012:**
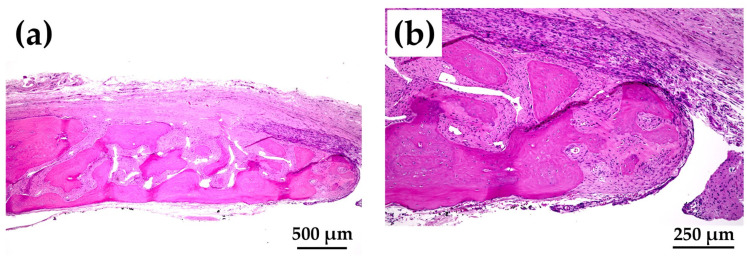
(**a**) Magnified HE-stained image of the dotted square zone “X1” in Figure 11c (4× magnification). (**b**) Further magnified HE-stained image of the bone edge in Figure 12a (10× magnification).

**Figure 13 materials-15-08802-f013:**
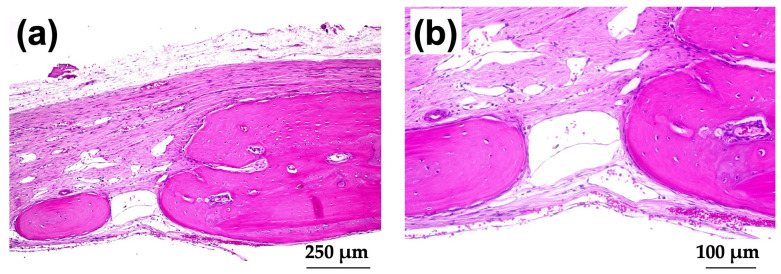
(**a**) Magnified HE-stained image of the dotted square zone “X2” in Figure 11d (4× magnification). (**b**) Further magnified HE-stained image of the interspace between the two bones in Figure 13a (10× magnification).

**Figure 14 materials-15-08802-f014:**
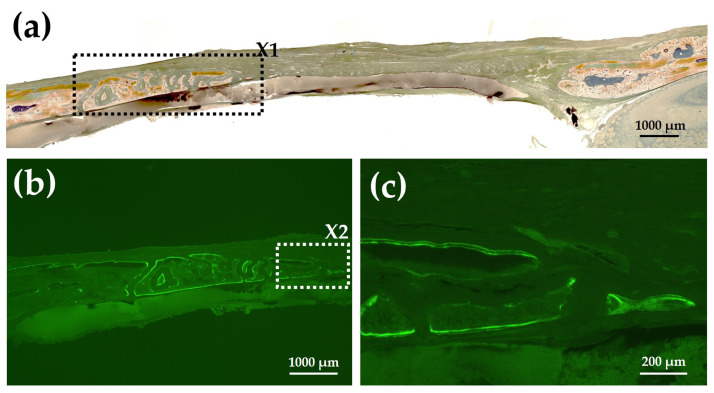
(**a**) Villanueva-stained image of a rat cranial bone defect filled with AG-infiltrated and b-FGF-loaded control collagen (Col control + AG + b-FGF) (4× magnification). (**b**) Magnified CL-fluor-labeled image corresponding to “X1” in Figure 14a (4× magnification). (**c**) Further magnified CL-fluor-labeled image corresponding to “X2” in Figure 14b (10× magnification).

**Figure 15 materials-15-08802-f015:**
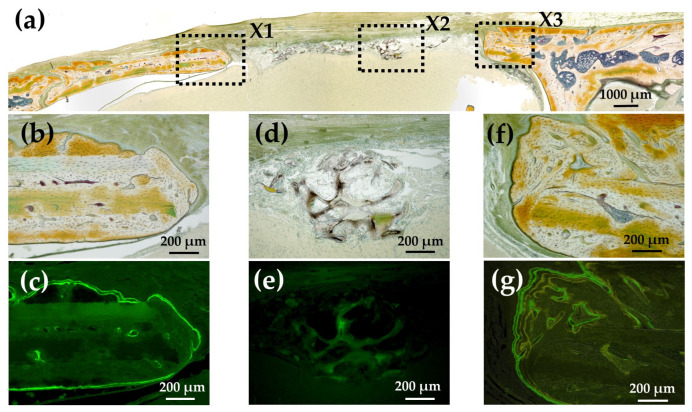
(**a**) Villanueva-stained image of a rat cranial bone defect filled with an AG-infiltrated and b-FGF-loaded Col/Hap composite (AI 20 min 5Cy Col/Hap + AG + b-FGF) (4× magnification). (**b**) Magnified image of “X1” in (**a**). (**c**) CL-fluor-labeled images of (**b**); (**d**) magnified image of “X2” in (**a**). (**e**) TC&CL-fluor-labeled image of (**d**). (**f**) Magnified image of “X3” in (**a**). (**g**) TC&CL-fluor-labeled image of (**f**) (10× magnification).

**Figure 16 materials-15-08802-f016:**
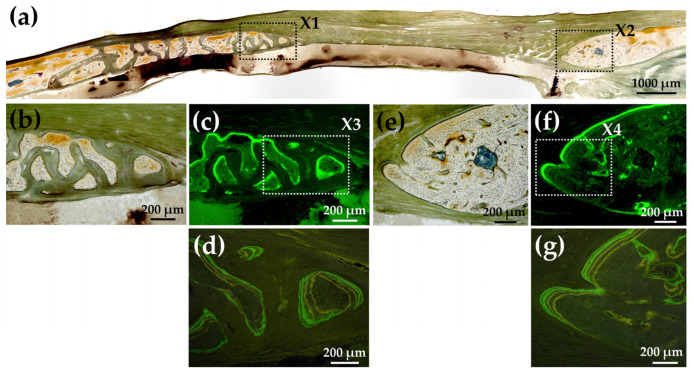
(**a**) Villanueva-stained image of a rat cranial bone defect filled with another AG-infiltrated and b-FGF-loaded Col/Hap composite (AI 60 min 5Cy Col/Hap + AG + b-FGF) (4× magnification). (**b**) Magnified image of “X1” in (**a**). (**c**) CL-fluor-labeled image of (**b**) (10× magnification). (**d**) Magnified TC&CL-fluor-labeled image of “X3” in (**c**) (20× magnification). (**e**) Magnified image of “X2” in (**a**). (**f**) CL-fluor-labeled image of (**e**) (10× magnification). (**g**) Magnified TC&CL-fluor-labeled image of “X4” in (**f**) (20× magnification).

**Figure 17 materials-15-08802-f017:**
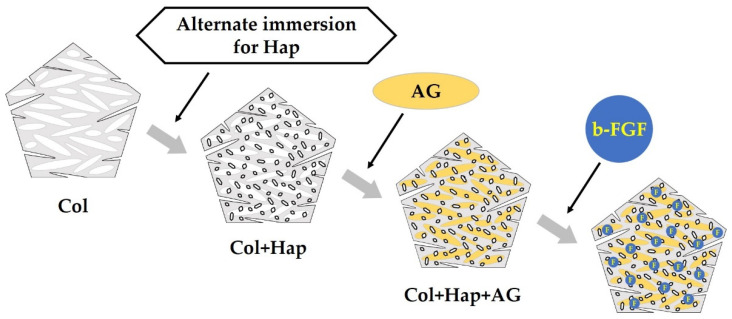
Preparation order and method for constructing granules consisting of collagen (Col), hydroxyapatite (Hap), acidic gelatin (AG), and b-FGF.

**Table 1 materials-15-08802-t001:** Code, composition, and preparation process (major part) of samples examined.

Code	Composition	Preparation Process (Major Part)
Before AG infiltration		
Col control	Col (Medical collagen)	24 h DHT treatment
AI 20 min 5Cy Col/Hap	Col, Hap	Alternate immersion of Col control in Ca^2+^ and PO_4_^−^ solutions for 20 min, respectively, 5 cycles
AI 60 min 5Cy Col/Hap	Col, Hap	Alternate immersion of Col control in Ca^2+^ and PO_4_^−^ solutions for 60 min, respectively, 5 cycles
After AG infiltration		
Col control + AG	Col, AG	Filling pores of Col control with AG
AI 20 min 5Cy Col/Hap + AG	Col, Hap, AG	Filling pores of AI 20 min 5Cy Col/Hap with AG
AI 60 min 5Cy Col/Hap + AG	Col, Hap, AG	Filling pores of AI 60 min 5Cy Col/Hap with AG
After impregnation of b-FGF		
Col control + AG + b-FGF	Col, AG, b-FGF	Dipping Col control + AG in b-FGF solution
AI 20 min 5Cy Col/Hap + AG + b-FGF	Col, Hap, AG, b-FGF	Dipping AI 20 min 5Cy Col/Hap + AG in b-FGF solution
AI 60 min 5Cy Col/Hap + AG + b-FGF	Col, Hap, AG, b-FGF	Dipping AI 60 min 5Cy Col/Hap + AG in b-FGF solution
Col/Hap/AG/b-FGF	Col, Hap, AG, b-FGF	Dipping Col/Hap/AG composite in b-FGF solution

**Table 2 materials-15-08802-t002:** Roles of components of the prepared granules (AI Col/Hap/AG) and b-FGF contained therein.

Materials Drug	Osteo- Conduction	Angiogenesis	b-FGF Loading and Slow Release	Remarks
Col	None	None	Minimum	Space-making
Hap	Large	Small	Small	Replaced to bone
AG	None	None	Large	Carrier of b-FGF
b-FGF	Small	Large	---	Wound healing Increase of stem cells

## Data Availability

All data are included in the manuscript.

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
