# Peer review of "Preparation of Collagen/Hydroxyapatite Composites Using the Alternate Immersion Method and Evaluation of the Cranial Bone-Forming Capability of Composites Complexed with Acidic Gelatin and b-FGF"

_materials, 2022, doi:10.3390/ma15248802_

Round 1
Reviewer 1 Report
The present study proposes the fabrication of a porous scaffold from collagen/apatite/acidic gelatin/basic fibroblast growth factor and evaluation of the cranial bone-forming capability. The work is impressive, its strong point being the in-vivo tests. However, the general impression was that the authors have a “template” for this kind of article, previously publishing 3 similar works in which only a few parameters varied. In this case, the only „innovation” seems to be the addition of the growth factor, as stated in the Introduction. Please find below my point-by-pint comments:
1. Abstract: “Quaternary constructs can be used as noble bone substitutes in the future.” - This is vague, please rephrase.
2. Keywords: please keep only the relevant ones for your work (e.g., delete the characterization techniques)
3. The Introduction contains too many details (e.g., about collagen and its crosslinking process), which are both well-known and irrelevant to the aim of this manuscript, which is the development of a new collagen/apatite/acidic gelatin (AG)/basic fibroblast growth factor (b-FGF) composites. The authors should rather focus on presenting the state-of-the-art and setting out the rationale behind the design of this work. An introduction should be a cohesive section to describe the topic (which is adequately represented in the first paragraph), followed by a brief review of the most recent bibliography, the rationale, the description of the actual work with a focus on its novelty and significance and finally the outline of the paper’s contents. In my opinion, it should be revised accordingly.
4. Materials and Methods: it is not clear why the authors presented the production of the collagenous membrane if further on they are not used (but collagen pellets) to prepare the final composite sponges. It is hard to follow the number and name notation for each sample, maybe a table with centralized synthesis parameters/differences would help.
5. L 137: the pH was not maintained at 7 using the equipment, but it was constantly monitored and corrected maybe by adding an acid/base; please check.
6. L140: how did the authors check if the crosslinker was entirely removed?
7. Sections 2.3.1 and 2.3.4 should be merged
8. Results: It is only now clear that the secondary scope of this manuscript was to produce and compare produced cross-linked collagen membranes with commercial ones. The novelty/importance of these aspects is not high and should be thoroughly described in the Introduction, mainly because the authors had similar previous works.
Author Response
Thank you for your advice and comment. I send you the Word file to answer your comments.

Reviewer 2 Report
Paper describes the preparation of collagen/apatite/acidic gelatin (AG)/basic fibroblast growth factor (b-FGF) composites with enhanced bone-forming capability. The work is of considerable interest for the development of a new generation of bone substitutes. However, the manuscript lacks more explanations and discussions and needs other improvements to be paper that is more valuable:
1) Figure 10 shows three curves (TG, DTA, DTG). However, only 2 curves (TG, DTA) are indicated in the figure caption.
2) Figure 13 shows the functional groups "CO" and "NH". However, other groups are given in the text (Section 3.2.3). In addition, there is no analysis of IR spectra. It is not clear how apatite precipitates (due to chemical or absorption interaction).
3) In Figure 15, along the abscissa axis, the authors incorrectly indicated the designation of the samples (“20 min” and “60 min” describe the conditions under which the samples were obtained, and not the samples themselves). In addition, the absolute values of the mass of absorbed b-FGF are given in the text, which makes it difficult to analysis the results obtained. It is necessary to give the relative values of the mass (to the mass or surface area of the original samples). In the caption to Figure 15, the number of samples should also be added during statistical processing of the results, as was done in Figure 17.
4) The term “hydroxy-apatite” is often used in this work. However, the common name for this calcium phosphate is “hydroxyapatite.”
5) The authors use many abbreviations, so it is necessary to add a decoding for the term “HE” to the Abbreviations.

Author Response
Thank you for your comments and advice. I send you a Word file to answer your comments.

Round 2
Reviewer 1 Report
The authors responded properly to my comments.